# The Associations of Spirituality, Adversity Quotient and Ethical Decision Making of Accounting Managers in the Contexts of Financial Management and Corporate Social Responsibility

**Hok-Ko Pong [1,\*] and Chun-Cheong Fong [2]**

1    Faculty of Management and Hospitality, Technology and Higher Education Institute of Hong Kong, Hong Kong, China
2    Faculty of Business, Macao Polytechnic University, Macau, China; ccfong@mpu.edu.mo
\*    Correspondence: hkpong@thei.edu.hk

**Abstract:** The objectives of this study are to explore the associations and interactions of spirituality, the adversity quotient (AQ), and the ethical decision making (EDM) of accounting managers in the contexts of financial management and corporate social responsibility. Additionally, the study aims to evaluate the predictive roles of spirituality and the adversity quotient (AQ) on their ethical decision making. A self-administered questionnaire was utilised to collect data from 510 accounting managers via the quantitative approach. The research results provide empirical evidence that the spiritual wellbeing (SWB) and AQ of accounting managers are positively correlated with EDM. In particular, the personal–communal domain of SWB is a significant predictor of moral equity, contractualism, egoism and deontology, whilst the environmental domain of SWB is a significant predictor of contractualism, utilitarianism, egoism and deontology. In terms of demographics, religion is a significant predictor of moral equity and deontology, whilst the origin and ownership dimension of AQ is a significant predictor of moral equity, contractualism, egoism and deontology. The control and reach dimensions of AQ are a significant predictor of moral equity respectively. Overall, individual characteristics, personal values, beliefs, interpersonal relationships and the environmental domain of SWB significantly influence EDM among accounting managers.

**Keywords:** spirituality; spiritual well-being; adversity quotient; ethical decision making; financial management; corporate social responsibility

## 1. Introduction

Accounting managers, such as chief financial officers and financial controllers, hold key positions in companies and are relevant players in the fields of financial management and corporate social responsibility (CSR), where ethical decision making (EDM) is important for corporate success. EDM helps to ensure the responsible and sustainable management of financial resources, accurate financial reporting and compliance with the governing laws and regulations of firms. Furthermore, EDM aligns financial management with organisational objectives, subsequently promoting long-term success and enhancing organisational reputation whilst preventing financial scandals, such as those encountered by Enron, WorldCom and Lehman Brothers.

From a legal perspective, managers, especially the accounting manager, must comply with a series of legal responsibilities and obligations. These managers must adhere to legal frameworks, such as the Sarbanes–Oxley Act in the US [1] or the Corporations Act in the UK [2]. These legal frameworks require accounting managers to maintain accurate financial records, implement effective internal controls and adhere to generally accepted accounting principles (GAAP) or International Financial Reporting Standards (IFRS) [3].

Accounting managers must also make sure that their companies adhere to financial reporting standards, pay the proper taxes on time, and meet audit requirements [4].

For instance, under the Sarbanes–Oxley Act, accounting managers are held personally accountable for inaccuracies in the financial reports of their companies [5]. This responsibility underscores the need for them to implement stringent internal controls and audit procedures. Also, accounting managers are expected to abide by ethical standards. Embezzlement, fraudulent financial reporting and insider trading are all crimes punishable by law [6]. For example, the Foreign Corrupt Practices Act (FCPA) in the United States provides for punishment of accounting managers if they fail to maintain accurate and transparent accounting records or if they engage in bribery of foreign officials [7].

Furthermore, accounting managers play a key role in maintaining corporate governance. They provide critical financial information for board decisions and participate in strategic planning and risk management [8]. In this regard, they are subject to various corporate governance laws and fiduciary duties. They are expected to comply with financial laws and regulations, uphold ethical standards, ensure tax compliance, and contribute to effective corporate governance. Their role is not only critical to business, but also legally affecting personal liability and corporate reputation [9]. However, business executives often face complex moral dilemmas that require them to be situated in challenging EDM scenarios, in which consequences impact not only their organisation but also society.

In recent years, the attention of professionals, practitioners and academicians has shifted increasingly toward determining how psychological and spiritual aspects influence the manner by which business professionals experience EDM. The adversity quotient (AQ) and spirituality are two aspects that have drawn particular attention.

In general, the capacity of a person to successfully deal with and handle difficult situations is referred to as their AQ. At work, AQ is positively correlated with EDM [10]. Meanwhile, spirituality covers a broad range of viewpoints and behaviours that may or may not be related to transcendental or divine intentions [11]. A previous study [12] established that spirituality is positively correlated with ethical conduct (e.g., EDM) in various business situations.

To gain new insights, deeper perspectives and a clear understanding of the factors that influence the decision-making process of accounting managers, it is valuable and meaningful to examine the interplay between spirituality, adversity quotient (AQ), and ethical decision making in the context of financial management and CSR.

The factors interacting to influence the EDM of firm managers should be thoroughly explored, especially in the financial management and CSR contexts, amid the growing interest in AQ, spirituality and EDM and their relationships.

This study aims to bridge existing research gaps by examining the interactions amongst the AQ, spiritual wellbeing (SWB) and EDM of accounting managers in financial management and CSR contexts. The primary goals of the study are to explore the relationships between SWB domains and EDM tendencies, examine the connections between AQ dimensions and EDM tendencies, assess the predictive power of SWB domains on the EDM tendencies of accounting managers, and evaluate the predictive ability of AQ dimensions on the EDM tendencies of accounting managers. To fulfil these goals, the study is guided by the following research questions:

Research Question 1: What constitutes the relationship between the SWB domains (personal–communal, environmental and transcendental domains) and EDM tendencies (moral equity, relativism, contractualism, utilitarianism, egoism and deontology) of accounting managers?
Research Question 2: What constitutes the relationship between the AQ dimensions (control, origin, ownership, reach and endurance dimensions) and EDM tendencies (moral equity, relativism, contractualism, utilitarianism, egoism and deontology) of accounting managers?
Research Question 3: Can SWB domains predict the EDM tendencies of accounting managers?
Research Question 4: Can AQ dimensions predict the EDM tendencies of accounting managers?

The theoretical framework of this research is based on a variety of theories and concepts, allowing for the illumination of the interactions of spirituality, AQ and EDM. Social

cognitive theory [13] dictates that individual characteristics, personal values and beliefs and interpersonal relationships significantly influence moral behaviour and conduct, including decision making. In the theory of SWB [14], spirituality is considered a multilayered concept encompassing personal–communal, environmental and transcendental domains. Study [15] determined the existence of a positive relationship between spirituality and the ethical judgment of business professionals. Similarly, study [16] established the EDM–AQ relationship by adopting the AQ theory [17], which proposes that the ability of a person to cope with and overcome challenging situations is an important component of success and ethical decision making in firms.

In EDM theory [18], EDM is assumed to involve a cognitive process of actualising reasoning and moral judgments based on ethical principles. A joint application of the aforementioned theories provides a framework for understanding how the spirituality, AQ and EDM constructs interact and subsequently influence each other.

## 2. Literature Review

### 2.1. AQ: Definition, Components and Measurement

Adversity is an unfavourable event or situation characterised by persistent and extreme difficulty. AQ, a relatively new concept in the psychology field, attempts to measure how individuals perceive and respond to adversity [17]. The four components of AQ are control, ownership, reach and endurance. Control refers to the degree to which individuals believe that they can influence events or situations. Ownership describes the degree to which a person feels personally responsible for their actions and outcomes. Reach is the ability and readiness of a person to search for and experience new challenges. Endurance pertains to the capacity of an individual to persevere under stress. The studies found that emotional intelligence, psychological wellbeing and professional performance are all positively correlated with AQ [19,20]. In the EDM setting, AQ is positively correlated with ethical behaviour and intentions [10]. These attributions suggest that AQ may encourage corporate executives to act morally.

### 2.2. Spirituality: Definition, Dimensions and Measurement

The broad and diversified concept of spirituality covers a wide range of beliefs and actions related to transcendental or divine intentions [21]. Transcendence—or finding meaning and purpose in life—and existential wellbeing are only a few of the various components of spirituality [22]. A common practice when measuring spirituality is to utilise self-reporting questionnaires to assess the different facets of this concept [23,24].

A variety of beneficial outcomes, including positive wellbeing [25], work satisfaction [26] and ethical conduct [26,27], have been positively associated with spirituality. As a significant predictor of EDM, moral identity is also positively correlated with spirituality [28]. Thus, spirituality encourages business professionals to make moral decisions.

### 2.3. EDM: Definition, Models and Influencing Factors

The EDM process is complicated because many ethical principles, values and norms need to be considered in a particular situation [29]. A variety of EDM models have been proposed, including the cognitive moral growth model [30] and social cognitive theory of moral thought and behaviour [31]. According to these theories, ethical judgments are a consequence of complex interactions between cognitive, emotional and behavioural processes [32].

Several influencing factors of EDM, including individual differences (moral identity and personality traits), situational factors (organisational culture and social norms) and external pressures (financial incentives and peer pressure), have been proposed [33,34]. Nevertheless, the process of selecting an ethical course of action remains challenging because of the intricate relationship of social, environmental and personal factors [35].

## 2.4. Previous Research on AQ and EDM

Despite the limited research on the AQ–EDM relationship, a positive relationship between AQ and ethical behaviour or intentions has been established [10]. Study [36] found that individuals with higher AQ are more likely to act morally even when confronted with difficult moral problems [36]. Furthermore, individuals with greater AQ may be more flexible in their thinking process and have relatively strong problem-solving abilities [37], further allowing them overcoming of ethical challenges in a more sophisticated and considerate manner. Theoretically, persons with greater AQ have higher levels of emotional intelligence and empathy, enabling them to better examine the implications of certain decisions [38]. However, the degree to which this relationship manifests is rarely studied.

In the CSR context, AQ can help individuals build a feeling of moral duty that is necessary for actualising EDM [10]. Consequently, organisations working with individuals with higher AQ are better equipped to execute their CSR obligations and can better contribute to an equitable and sustainable global community [39]. Furthermore, people with high AQ are more conscious of how their financial decisions impact stakeholders and the greater community; in this sense, AQ may promote ethical conduct in financial management. People are more responsible when they are equipped to decide on matters that are consistent with their moral values, even when certain decisions need to be compromised [10]. These scenarios suggest that AQ promotes ethical conduct and intentions even in the face of difficult ethical dilemmas. However, the underlying interacting mechanisms of AQ and moral judgment remain poorly explored, especially in the financial management and CSR contexts.

## 2.5. Extant Research on Spirituality and EDM

In the CSR context, spirituality helps individuals make moral choices. Moral responsibility [40], which may be nurtured via spirituality, is a requirement for ethical action in CSR. Study [41] suggests that spirituality fosters ethical behaviour in financial management. For example, amongst Islamic bankers required to arrive at financial decisions, spirituality is positively correlated with moral conduct. Spirituality also encourages social justice and a sense of interconnectedness with stakeholders, both of which are important components of moral conduct [28]. Similarly, spirituality promotes the ethical behaviour (e.g., EDM) of powerful and influential individuals, especially in business settings, allowing them avoidance of moral dilemmas that may eventually lead to extensive repercussions.

## 2.6. Theoretical Framework: The Influences and Interaction of AQ and SWB on EDM

Theoretically, certain factors interact to affect EDM. However, the influence and interaction of AQ and SWB on EDM is rarely examined. Self-awareness and confidence, which are examples of the capacity of individuals to overcome obstacles, are elements of AQ and spirituality, respectively. A stronger sense of moral identity may result from a shared sense of self-awareness and resiliency, which may subsequently encourage moral decision making [36]. Furthermore, spirituality affords individuals a sense of direction and meaning, which may increase their drive to behave morally [42].

Overall, the reviewed literature emphasises the importance of investigating the influence of psychological and spiritual factors on the EDM of business professionals. Theoretically, certain factors interact to encourage ethical behaviours and intentions. AQ and spirituality have both been positively associated with EDM in various contexts. However, the mechanisms underlying the relationships and how they function specifically in the financial management and CSR contexts remain largely unexplored.

Based on the theoretical framework and previous studies, the four hypotheses adopted in this study are as follows:

**Hypothesis 1 (H1):** *Amongst accounting managers, a positive correlation exists between SWB (personal–communal, environmental and transcendental domains) and ethical judgments for decision making (moral equity, relativism, contractualism, utilitarianism, egoism and deontology). H1 is*

*based on the findings of study* [15], *which established the relationship between spirituality and ethical judgment amongst business professionals.*

**Hypothesis 2 (H2):** *AQ is positively correlated with moral judgment and morally superior choices. H2 is supported by study* [16], *which established the interaction between EDM and AQ.*

**Hypothesis 3 (H3):** *Amongst accounting managers, SWB is a significant predictor of moral judgment. In particular, the ethical judgments of accounting managers may be influenced by their spirituality.*

**Hypothesis 4 (H4):** *Amongst accounting managers, AQ is a significant predictor of moral judgment. In particular, AQ may help to predict the ethical and moral judgment of accounting managers.*

## 3. Methodology

This study used three instruments, namely (1) the Spiritual Health and Life Orientation Measure (SHALOM) [23], (2) the Adversity Response Profile (ARP) [17] and (3) the Multidimensional Ethics Scale (MES) [18], to explore the association and interactions of SWB, AQ and EDM of accounting managers engaged in financial management and CSR. The data were collected using a questionnaire developed on the basis of a literature review and consultation with field experts. The questionnaire covered four segments. In the first segment, data on the age, gender, level of education and years of work experience of participants were collected. The remaining three portions of the tool are discussed in the subsequent sections.

### 3.1. SWB

SHALOM was used in the second segment to assess the SWB of the respondents. Twenty components comprised the tool and were then distributed into four domains: personal, communal, environmental and transcendental. The participants were instructed to formulate two answers to each question: one based on their own experience and the other based on their ideal values. Then, these responses were integrated as components of the SWB questionnaire (i.e., a questionnaire utilised in this study). Each reply was scored between 1 and 5 (1 = 'strongly disagree'; 5 = 'strongly agree'). Regarding the merits of using the SWB questionnaire, more than 30 languages have been used to translate its 20 items, and the tool has consistently shown good dependability in many studies.

In this study, a three-factor SWB questionnaire framework was utilised. The personal domain was combined with the communal domain to produce a single category, taking into account the Confucian ideals central to Chinese culture and family unity and harmonious connections with others. The concepts of God or Creator, ancestors, a higher power and the higher self were also adopted [25] in the transcendental section of the SWB questionnaire.

### 3.2. AQ

The ARP was used in the third segment to assess the AQ of the research subjects. The respondents were asked to rate their feelings toward 20 scenarios, in which a five-point Likert scale (1 = 'not at all'; 5 = 'completely') was used for scoring. Four AQ components were evaluated: control (five items), origin and ownership (five items), reach (five items) and endurance (five items).

### 3.3. EDM

The MES tool [18] was used in the fourth segment to assess the degree to which the participants consider EDM to be effective. The MES included 30 items, corresponding to five items for each of the six EDM characteristics (moral equity, relativism, contractualism, utilitarianism, egoism and deontology) developed by the authors for this study. For each question, the respondents were instructed to rank the morality of a fictitious activity on a

scale of 1 to 7 (1 = 'morally unacceptable'; 7 = 'morally acceptable'). The task items were specifically created to gauge the intensity of moral judgment that each participant attributes to each of the six ethical qualities.

The questionnaire items used in this study were slightly modified to suit financial management and CSR. A sample item is 'This action is consistent with fulfilling moral duties and obligations in financial management and corporate social responsibility'.

Prior to administering the final version of the questionnaire to 1150 accounting practitioners in Hong Kong, a pre-test was conducted amongst a group of accounting practitioners to ensure that the items were clear and relevant. The participants were randomly selected from a list provided by the Hong Kong Institute of Certified Public Accountants. Both online and paper formats were offered to the participants, allowing them the choice of their preferred questionnaire format. A total of 510 responses were received, corresponding to a response rate of 44.3%. The data collection period spanned October 2022 to March 2023. The data were collected for analysis in April 2023.

The research methodology was executed following the guidelines of the Ethics Committee of the authors' institution, thus ensuring the ethical conduct of the research process. Research subjects who confirmed their consent to participate in the research were fully informed of the research purpose and objectives; they were also guaranteed confidentiality and anonymity. All of the collected data were securely stored and were accessible only to the research team. These data were exclusively used for this particular research and were not shared with any third party. Furthermore, the respondents were provided with the contact information of the researchers should they have any questions or concerns. The ethical application and procedures of this research prioritised the protection of the rights and welfare of the respondents whilst maintaining the integrity and credibility of the research findings.

## 4. Results

Confirmatory factor analysis (CFA) was used to examine the factor structures of the three questionnaires (SWB questionnaire, ARP and MES) and assess the model fit, consequently providing evidence of the validity of the measurement tools. The three-factor structure of the SWB questionnaire showed a good fit with the data, with CFI = 0.928, TLI = 0.918, SRMR = 0.0578 and RMSEA = 0.0737. Similarly, the four-factor structure of the ARP also showed a good fit with the data, with CFI = 0.913, TLI = 0.899, SRMR = 0.0903 and RMSEA = 0.109. A good fit with the data was also found for the six-factor structure of the MES, with CFI = 0.953, TLI = 0.948, SRMR = 0.0393 and RMSEA = 0.0767.

Then, the internal consistency of the SWB questionnaire was assessed using Cronbach's $\alpha$ for each of the SWB domains. The Cronbach's $\alpha$ values of the personal–communal, environmental and transcendental domains were 0.907, 0.828 and 0.950, respectively. The full-scale SWB questionnaire had a Cronbach's $\alpha$ of 0.914, indicating good internal consistency.

The ARP and MES measurement tools were also evaluated for internal consistency. For the ARP tool, the Cronbach's $\alpha$ values for the control, origin, ownership, reach and endurance domains were 0.824, 0.955, 0.912 and 0.902, respectively. The entire ARP scale obtained a Cronbach's $\alpha$ of 0.905, suggesting strong internal consistency. For the MES tool, the Cronbach's $\alpha$ values for the moral equity, relativism, contractualism, utilitarianism, egoism and deontology variables were 0.940, 0.909, 0.969, 0.928 and 0.981, respectively. The Cronbach's $\alpha$ for the entire MES was 0.865, which denotes favourable internal consistency.

The results of the internal consistency analysis and reliability test showed that the measurement tools (i.e., the SWB questionnaire, ARP and MES) can reliably process indicators of SWB, adversity resistance and EDM processes, respectively. These results allowed the researchers the subsequent performance of regression analyses to sufficiently evaluate H1, which seeks to investigate the positive relationship between SWB and AQ. H2 attempts to determine whether the MES tool can predict the relationship between SWB and AQ. The findings can further provide insights into the interactions of SWB and AQ and the ways in

which they can predict the decision-making abilities of accounting managers in financial management and CSR contexts.

## 4.1. Descriptive Results

Table 1 summarises the descriptive data obtained for this study. It highlights the characteristics of the participants and the substantial variations in the MES across demographic variables. A total of 510 participants were recruited in this research, with 44.3% men and 55.7% women. The ages of the participants ranged from <25 to ≥49. The 25–31 age group represented the largest portion of participants (32.9%), followed by the 32–39 age group (27.1%). The respondents who were 49 years of age or older comprised the smallest age group (11.2%).

**Table 1.** Accounting Manager (N = 510) Characteristics and Scores on the MES.

| | | N (%) | Moral Equity (MES) | Relativism (MES) | Contractualism (MES) | Utilitarianism (MES) | Egoism (MES) | Deontology (MES) | Overall (MES) |
|---|---|---|---|---|---|---|---|---|---|
| | | | M (S.D.) | M (S.D.) | M (S.D.) | M (S.D.) | M (S.D.) | M (S.D.) | M (S.D.) |
| All | | 510 (100%) | 4.63 (0.52) | 5.67 (0.56) | 5.04 (0.66) | 4.79 (0.61) | 4.18 (0.91) | 4.02 (0.78) | 4.72 (0.34) |
| Gender | (1) Male | 226 (44.3%) | 4.54 (0.56) | 5.64 (0.58) | 5.08 (0.64) | 4.78 (0.57) | 4.18 (0.92) | 3.92 (0.74) | 4.69 (0.34) |
| | (2) Female | 284 (55.7%) | 4.70 (0.48) | 5.70 (0.55) | 5.01 (0.66) | 4.80 (0.63) | 4.17 (0.91) | 4.10 (0.80) | 4.75 (0.33) |
| | | | $t = -3.398$ ** | $t = -1.352$ | $t = 1.149$ | $t = -0.350$ | $t = 0.196$ | $t = -2.543$ | $t = -1.861$ |
| Age | (1) Under 25 years old | 63 (12.4%) | 4.54 (0.57) | 5.60 (0.54) | 5.16 (0.54) | 4.87 (0.56) | 4.11 (0.95) | 4.02 (0.82) | 4.72 (0.37) |
| | (2) 25–31 years old | 168 (32.9%) | 4.54 (0.56) | 5.62 (0.56) | 5.09 (0.63) | 4.75 (0.58) | 4.19 (0.92) | 4.04 (0.77) | 4.71 (0.32) |
| | (3) 32–39 years old | 138 (27.1%) | 4.64 (0.51) | 5.71 (0.57) | 4.94 (0.68) | 4.79 (0.62) | 4.13 (0.89) | 3.97 (0.82) | 4.70 (0.33) |
| | (4) 40–48 years old | 84 (16.5%) | 4.73 (0.45) | 5.75 (0.62) | 5.00 (0.70) | 4.73 (0.64) | 4.20 (0.81) | 3.96 (0.79) | 4.73 (0.35) |
| | (5) 49 years old or older | 57 (11.2%) | 4.79 (0.43) | 5.72 (0.50) | 5.09 (0.61) | 4.93 (0.63) | 4.27 (1.04) | 4.14 (0.68) | 4.82 (0.36) |
| | | | $F = 3.937$ ** | $F = 1.329$ | $F = 1.752$ | $F = 1.461$ | $F = 0.339$ | $F = 0.580$ | $F = 1.572$ |
| Post Hoc Analysis (LSD) | | | (4) & (5) > (1) & (2) | | | | | | |
| Education Level | (1) Diploma/ associate degree | 82 (16.1%) | 4.69 (0.52) | 5.66 (0.52) | 5.09 (0.76) | 4.86 (0.64) | 4.31 (0.96) | 4.10 (0.77) | 4.79 (0.35) |
| | (2) Bachelor's degree | 307 (60.2%) | 4.64 (0.52) | 5.68 (0.56) | 5.02 (0.64) | 4.81 (0.57) | 4.16 (0.88) | 3.99 (0.77) | 4.72 (0.33) |
| | (3) Postgraduate degree | 121 (23.7%) | 4.56 (0.54) | 5.68 (0.60) | 5.05 (0.62) | 4.69 (0.67) | 4.14 (0.94) | 4.04 (0.81) | 4.69 (0.34) |
| | | | $F = 1.509$ | $F = 0.027$ | $F = 0.386$ | $F = 2.337$ | $F = 1.107$ | $F = 0.757$ | $F = 1.953$ |

Table 1. *Cont.*

|  |  | N (%) | Moral Equity (MES) | Relativism (MES) | Contractualism (MES) | Utilitarianism (MES) | Egoism (MES) | Deontology (MES) | Overall (MES) |
|---|---|---|---|---|---|---|---|---|---|
| Work experience | (1) Less than 3 years | 65 (12.7%) | 4.54 (0.57) | 5.58 (0.54) | 5.17 (0.65) | 4.87 (0.57) | 4.13 (0.94) | 4.03 (0.83) | 4.72 (0.38) |
|  | (2) 3–8 years | 172 (33.7%) | 4.55 (0.55) | 5.63 (0.57) | 5.08 (0.62) | 4.76 (0.58) | 4.19 (0.93) | 4.04 (0.76) | 4.71 (0.31) |
|  | (3) 9–17 years | 157 (30.8%) | 4.64 (0.52) | 5.69 (0.55) | 4.93 (0.66) | 4.82 (0.61) | 4.15 (0.89) | 3.94 (0.79) | 4.70 (0.33) |
|  | (4) 18–26 years | 86 (16.9%) | 4.79 (0.38) | 5.77 (0.63) | 5.01 (0.72) | 4.70 (0.61) | 4.17 (0.87) | 4.02 (0.80) | 4.74 (0.35) |
|  | (5) 27 years or older | 30 (5.9%) | 4.72 (0.53) | 5.73 (0.45) | 5.15 (0.63) | 4.98 (0.75) | 4.37 (0.99) | 4.24 (0.68) | 4.87 (0.40) |
|  |  |  | $F = 3.970$ ** | $F = 1.306$ | $F = 2.113$ | $F = 1.781$ | $F = 0.428$ | $F = 0.999$ | $F = 1.722$ |
| Post Hoc Analysis (LSD) |  |  | (1) and (2) < (3) and (4) |  |  |  |  |  |  |
| Salaries (Monthly) | (1) HKD 20,000–HKD 39,999 | 81 (15.9%) | 4.68 (0.52) | 5.67 (0.51) | 5.11 (0.76) | 4.87 (0.64) | 4.31 (0.97) | 4.10 (0.77) | 4.79 (0.35) |
|  | (2) HKD 40,000–HKD 59,999 | 295 (57.8%) | 4.63 (0.52) | 5.68 (0.56) | 5.01 (0.64) | 4.81 (0.57) | 4.13 (0.88) | 3.96 (0.76) | 4.70 (0.33) |
|  | (3) HKD 60,000–HKD 79,999 | 114 (22.4%) | 4.54 (0.55) | 5.65 (0.58) | 5.03 (0.62) | 4.68 (0.68) | 4.13 (0.95) | 3.98 (0.79) | 4.67 (0.33) |
|  | (4) HKD 80,000 or higher | 20 (3.9%) | 4.93 (0.16) | 5.82 (0.70) | 5.31 (0.65) | 4.92 (0.53) | 4.59 (0.72) | 4.88 (0.56) | 5.08 (0.26) |
|  |  |  | $F = 3.726$ | $F = 0.545$ | $F = 1.667$ | $F = 2.146$ | $F = 2.350$ | $F = 9.602$ *** | $F = 10.232$ *** |
| Post Hoc Analysis (LSD) |  |  | (4) > (2) and (3) |  |  |  |  | (4) > (1), (2) and (3) | (2) and (3) < (1) < (4) |
| Religious | (1) Not religious | 371 (62.2%) | 4.53 (0.53) | 5.67 (0.56) | 5.02 (0.69) | 4.80 (0.61) | 4.14 (0.92) | 3.86 (0.76) | 4.67 (0.31) |
|  | (2) Religious | 193 (37.8%) | 4.79 (0.47) | 5.69 (0.57) | 5.07 (0.60) | 4.78 (0.59) | 4.23 (0.90) | 4.28 (0.75) | 4.81 (0.37) |
|  |  |  | $t = -5.494$ *** | $t = -0.436$ | $t = -0.941$ | $t = 0.329$ | $t = -1.084$ | $t = -6.129$ *** | $t = -4.524$ *** |

Note: ***: $p < 0.001$; ** $p < 0.001$.

Nearly two thirds (60.2%) of the participants completed a bachelor's degree, followed by those with a postgraduate degree (23.7%) and a diploma/associate degree (16.1%). The majority of participants (33.7%) worked for 3–8 years, and the second largest group (30.8%) worked for 9–17 years. Amongst the respondents, 5.9% were employed for 27 years or longer. Participants who earned between HKD 40,000 and HKD 59,999 (57.8%) comprised most of the respondents, followed by those who earned between HKD 60,000 and HKD 79,999 (22.4%), HKD 20,000 and HKD 39,999 (15.9%) and HKD 80,000 or higher (3.9%). In terms of religious affiliations, 62.2% of the research subjects claimed to not practice any religion, whereas 37.8% claimed to practice one.

The study used a *t* test and one-way ANOVA to determine how the characteristics of MES varied across different populations. On the basis of gender, age, job experience and monthly wage, statistically significant variations ($p < 0.001$) were observed for the MES variables (Table 1). In terms of moral justice, female accounting managers performed substantially better ($p < 0.001$) than male accounting managers, and older accounting managers performed significantly better ($p < 0.001$) than younger accounting managers. Post hoc analysis (LSD) also established that accounting managers aged at ≥40 years performed much better in terms of moral equity with respect to the MES variables than accounting managers who are <30 years old ($p < 0.001$). Accounting managers with more

work experience, particularly >18 years of work experience, scored significantly higher ($p < 0.001$) than those with less experience (i.e., <8 years). Monthly salary was a significant factor. In particular, managers with higher monthly salaries scored higher in terms of deontology in the MES tool. Similarly, for the deontology variable, post hoc analysis (LSD) revealed that accounting managers with a monthly salary of HKD 80,000 or higher scored significantly higher ($p < 0.001$) than those who earned less than HKD 80,000.

### 4.2. Pearson Correlation

The Pearson correlation coefficient ($r$) was used to examine the EDM–SWB relationship and EDM–AQ relationship of accounting managers. Table 2 shows a significant positive association across all three specific domains (personal–communal, environmental and transcendental domains) of SWB of accounting managers and the moral equity, contractualism, egoism and deontology variables of the MES tool.

**Table 2.** Pearson correlations between three domains of SWB and AQ and MES of Chinese accounting managers (N = 510).

| | Moral Equity (MES) | Relativism (MES) | Contractualism (MES) | Utilitarianism (MES) | Egoism (MES) | Deontology (MES) | Overall (MES) |
|---|---|---|---|---|---|---|---|
| Personal–communal (SWB) | 0.625 ** | 0.053 | 0.198 ** | 0.067 | 0.148 * | 0.506 ** | 0.519 ** |
| Environmental (SWB) | 0.300 ** | 0.047 | 0.180 ** | 0.113 * | 0.175 * | 0.362 ** | 0.399 ** |
| Transcendental (SWB) | 0.224 ** | 0.010 | 0.141 ** | 0.046 | 0.019 | 0.273 ** | 0.232 ** |
| Overall (SWB) | 0.512 ** | 0.046 | 0.221 ** | 0.090 * | 0.136 ** | 0.494 ** | 0.493 ** |
| Control (AQ) | 0.338 ** | 0.011 | 0.056 | −0.021 | 0.058 | 0.252 ** | 0.224 ** |
| Origin and Ownership (AQ) | 0.632 ** | 0.048 | 0.259 ** | 0.103 * | 0.190 ** | 0.523 ** | 0.576 ** |
| Reach (AQ) | 0.444 ** | 0.041 | −0.067 | −0.075 | 0.076 | 0.276 ** | 0.221 ** |
| Endurance (AQ) | 0.276 ** | 0.065 | 0.039 | −0.020 | 0.007 | 0.210 ** | 0.179 ** |
| Overall (AQ) | 0.638 ** | 0.060 | 0.122 ** | 0.008 | 0.135 ** | 0.480 ** | 0.467 ** |

Note: ** $p < 0.001$; * $p < 0.05$.

Table 2 also shows significant positive correlations across all three specific domains of SWB and moral equity (from $r = 0.224$ ** to $r = 0.625$ **), contractualism (from $r = 0.141$ ** to $r = 0.221$ **), egoism (from $r = 0.136$ ** to $r = 0.175$ *) and deontology (from $r = 0.273$ ** to $r = 0.506$ **). Moral equity showed the strongest positive correlation with SWB in the personal–communal domain ($r = 0.625$ **) and the overall domain ($r = 0.512$ **). These findings indicate that people with higher levels of SWB are more likely to actualise EDM based on the principles of fairness, justice and duty.

Deontology presented the strongest positive correlation with SWB in the personal–communal domain ($r = 0.506$ **) and the overall domain ($r = 0.494$ **). This finding suggests that people with higher levels of SWB are likely to prioritise moral rules and principles in their decision-making process. However, relativism (a variable of MES) was not significantly correlated with any of the specific SWB categories. Meanwhile, contractualism exhibited a significant positive relationship with SWB in all of its specific domains (from $r = 0.180$ ** to $r = 0.221$ **). Utilitarianism and egoism were significantly associated with SWB in the environmental domain (from $r = 0.113$ * to 0.175 *), although the correlation was relatively weak.

The results further showed significant positive correlations across all specific dimensions of AQ and the variables of moral equity (from $r = 0.276$ ** to $r = 0.638$ **), deontology (from $r = 0.210$ ** to $r = 0.523$ **) for MES and the overall score (from $r = 0.179$ ** to $r = 0.576$ **). In particular, moral equity had the strongest positive correlation with AQ in terms of origin and ownership ($r = 0.632$ **) and the overall domain ($r = 0.638$ **). The findings indicate that individuals who possess high levels of resilience and capacity to cope with challenging situations also possess a strong sense of fairness, justice and duty; they

likely rely on ethical decisions that align with the aforementioned principles. In contrast, contractualism (*r* = 0.259 **), utilitarianism (*r* = 0.103 *), egoism (*r* = 0.190 **) and deontology (*r* = 0.523 **) presented significant positive but weaker correlations with AQ in the origin and ownership dimension. No significant correlations were found between relativism (an MES variable) and any of the specific AQ dimensions.

The abovementioned results partially support H1, which suggests a positive correlation between (1) EDM and SWB and (2) between EDM and AQ amongst business professionals.

### 4.3. Hierarchical Regression Analysis

Tables 3–7 present the results of the hierarchical regression analyses conducted in this study. The specific SWB domains and AQ dimensions were used as predictors of EDM.

**Table 3.** Results of Hierarchical Regression Analyses with SWB and AQ as Predictors of Participant MES in moral equity.

| Variable | | | β | t | F | R | $R^2$ | $\Delta R^2$ | Adjusted $R^2$ |
|---|---|---|---|---|---|---|---|---|---|
| Moral equity | | | | | | | | | |
| Step 1 | | | | | 12.663 ** | 0.302 | 0.091 | 0.091 | 0.084 |
| | Demographics | | | | | | | | |
| | | Gender | 0.110 | 2.561 | | | | | |
| | | Age | 0.185 | 1.362 | | | | | |
| | | Work Experience | −0.043 | −0.319 | | | | | |
| | | Religion | 0.213 | 4.973 ** | | | | | |
| Step 2 | | | | | 68.480 ** | 0.636 | 0.405 | 0.313 | 0.399 |
| | Demographics | | | | | | | | |
| | | Gender | 0.027 | 0.780 | | | | | |
| | | Age | 0.108 | 0.983 | | | | | |
| | | Work Experience | −0.002 | −0.021 | | | | | |
| | | Religion | 0.038 | 1.051 | | | | | |
| | SWB: | | | | | | | | |
| | | Personal–communal | 0.598 | 16.286 ** | | | | | |
| Step 3 | | | | | 57.155 ** | 0.637 | 0.405 | 0.001 | 0.399 |
| | Demographics | | | | | | | | |
| | | Gender | 0.026 | 0.724 | | | | | |
| | | Age | 0.108 | 0.985 | | | | | |
| | | Work Experience | −0.003 | −0.026 | | | | | |
| | | Religion | 0.040 | 1.087 | | | | | |
| | SWB: | | | | | | | | |
| | | Personal–communal | 0.616 | 14.555 ** | | | | | |
| | | Environmental | −0.034 | −0.849 | | | | | |
| Step 4 | | | | | 48.959 ** | 0.637 | 0.406 | 0.000 | 0.399 |
| | Demographics | | | | | | | | |
| | | Gender | 0.024 | 0.687 | | | | | |
| | | Age | 0.108 | 0.976 | | | | | |
| | | Work Experience | −0.001 | −0.012 | | | | | |
| | | Religion | 0.038 | 1.046 | | | | | |

**Table 3.** *Cont.*

| Variable | | β | t | F | R | R² | ΔR² | Adjusted R² |
|---|---|---|---|---|---|---|---|---|
| | SWB: | | | | | | | |
| | Personal–communal | 0.612 | 14.284 ** | | | | | |
| | Environmental | −0.039 | −0.947 | | | | | |
| | Transcendental | 0.020 | 0.524 | | | | | |
| Step 5 | | | | 44.927 ** | 0.646 | 0.418 | 0.012 | 0.408 |
| | Demographics | | | | | | | |
| | Gender | 0.032 | 0.912 | | | | | |
| | Age | 0.130 | 1.191 | | | | | |
| | Work Experience | −0.022 | −0.204 | | | | | |
| | Religion | 0.028 | 0.775 | | | | | |
| | SWB: | | | | | | | |
| | Personal–communal | 0.568 | 12.721 ** | | | | | |
| | Environmental | −0.041 | −0.991 | | | | | |
| | Transcendental | 0.022 | 0.583 | | | | | |
| | AQ: | | | | | | | |
| | Control | 0.120 | 3.214 * | | | | | |
| Step 6 | | | | 49.999 ** | 0.688 | 0.474 | 0.056 | 0.464 |
| | Demographics | | | | | | | |
| | Gender | 0.023 | 0.695 | | | | | |
| | Age | 0.109 | 1.051 | | | | | |
| | Work Experience | −0.022 | −0.208 | | | | | |
| | Religion | −0.018 | −0.513 | | | | | |
| | SWB: | | | | | | | |
| | Personal–communal | 0.338 | 6.382 ** | | | | | |
| | Environmental | −0.060 | −1.539 | | | | | |
| | Transcendental | 0.006 | 0.171 | | | | | |
| | AQ: | | | | | | | |
| | Control | 0.106 | 2.981 * | | | | | |
| | Origin and ownership | 0.363 | 7.291 ** | | | | | |
| Step 7 | | | | 51.058 ** | 0.711 | 0.506 | 0.032 | 0.496 |
| | Demographics | | | | | | | |
| | Gender | 0.027 | 0.827 | | | | | |
| | Age | 0.096 | 0.945 | | | | | |
| | Work Experience | −0.017 | −0.165 | | | | | |
| | Religion | −0.035 | −1.022 | | | | | |
| | SWB: | | | | | | | |
| | Personal–communal | 0.238 | 4.398 ** | | | | | |
| | Environmental | −0.052 | −1.373 | | | | | |
| | Transcendental | −0.003 | −0.101 | | | | | |
| | AQ: | | | | | | | |
| | Control | 0.072 | 2.068 * | | | | | |
| | Origin and ownership | 0.392 | 8.076 ** | | | | | |
| | Reach | 0.207 | 5.689 ** | | | | | |

**Table 3.** *Cont.*

| Variable | | | β | t | F | R | R² | ΔR² | Adjusted R² |
|---|---|---|---|---|---|---|---|---|---|
| Step 8 | | | | | 46.427 ** | 0.712 | 0.506 | 0.001 | 0.495 |
| | Demographics | | | | | | | | |
| | | Gender | 0.027 | 0.837 | | | | | |
| | | Age | 0.097 | 0.958 | | | | | |
| | | Work Experience | −0.016 | −0.162 | | | | | |
| | | Religion | −0.034 | −0.981 | | | | | |
| | SWB: | | | | | | | | |
| | Personal–communal | | 0.240 | 4.423 ** | | | | | |
| | Environmental | | −0.052 | −1.359 | | | | | |
| | Transcendental | | −0.007 | −0.189 | | | | | |
| | AQ: | | | | | | | | |
| | Control | | 0.084 | 2.192 * | | | | | |
| | Origin and ownership | | 0.399 | 8.085 ** | | | | | |
| | Reach | | 0.205 | 5.642 ** | | | | | |
| | Endurance | | −0.029 | −0.750 | | | | | |

Note: ** $p < 0.001$; * $p < 0.05$.

**Table 4.** Results of Hierarchical Regression Analyses with SWB and AQ as Predictors of Participant MES in Contractualism.

| Variable | | β | t | F | R | R² | ΔR² | Adjusted R² |
|---|---|---|---|---|---|---|---|---|
| Contractualism | | | | | | | | |
| Step 1 | | | | 20.661 ** | 0.198 | 0.039 | 0.039 | 0.037 |
| | SWB: | | | | | | | |
| Personal–communal | | 0.198 | 4.545 ** | | | | | |
| Step 2 | | | | 12.576 ** | 0.217 | 0.047 | 0.008 | 0.044 |
| | SWB: | | | | | | | |
| Personal–communal | | 0.142 | 2.803 * | | | | | |
| Environmental | | 0.106 | 2.087 * | | | | | |
| Step 3 | | | | 9.016 ** | 0.225 | 0.051 | 0.003 | 0.045 |
| | SWB: | | | | | | | |
| Personal–communal | | 0.129 | 2.495 * | | | | | |
| Environmental | | 0.090 | 1.720 | | | | | |
| Transcendental | | 0.064 | 1.361 | | | | | |
| Step 4 | | | | 10.279 ** | 0.274 | 0.075 | 0.025 | 0.068 |
| | SWB: | | | | | | | |
| Personal–communal | | −0.034 | −0.501 | | | | | |
| Environmental | | 0.077 | 1.486 | | | | | |
| Transcendental | | 0.052 | 1.109 | | | | | |
| | AQ: | | | | | | | |
| Origin and ownership | | 0.235 | 3.662 ** | | | | | |

Note: ** $p < 0.001$; * $p < 0.05$.

**Table 5.** Results of Hierarchical Regression Analyses with SWB and AQ as Predictors of Participant MES in Utilitarianism.

| Variable | | | β | t | F | R | R$^2$ | ΔR$^2$ | Adjusted R$^2$ |
|---|---|---|---|---|---|---|---|---|---|
| Utilitarianism | | | | | | | | | |
| Step 1 | | | | | 6.578 * | 0.113 | 0.013 | 0.013 | 0.011 |
| | SWB: | | | | | | | | |
| | | Environmental | 0.113 | 2.565 * | | | | | |
| Step 2 | | | | | 4.224 * | 0.128 | 0.016 | 0.004 | 0.013 |
| | SWB: | | | | | | | | |
| | | Environmental | 0.084 | 1.715 | | | | | |
| | AQ: | | | | | | | | |
| | | Origin and ownership | 0.067 | 1.364 | | | | | |

Note: * $p < 0.05$.

**Table 6.** Results of Hierarchical Regression Analyses with SWB and AQ as Predictors of Participant MES in Egoism.

| Variable | | | β | t | F | R | R$^2$ | ΔR$^2$ | Adjusted R$^2$ |
|---|---|---|---|---|---|---|---|---|---|
| Egoism | | | | | | | | | |
| Step 1 | | | | | 11.318 ** | 0.148 | 0.022 | 0.022 | 0.020 |
| | SWB: | | | | | | | | |
| | | Personal–communal | 0.148 | 3.364 * | | | | | |
| Step 2 | | | | | 9.165 ** | 0.187 | 0.035 | 0.013 | 0.031 |
| | SWB: | | | | | | | | |
| | | Personal–communal | 0.078 | 1.519 | | | | | |
| | | Environmental | 0.134 | 2.623 * | | | | | |
| Step 3 | | | | | 8.306 ** | 0.217 | 0.047 | 0.012 | 0.041 |
| | SWB: | | | | | | | | |
| | | Personal–communal | −0.038 | −0.555 | | | | | |
| | | Environmental | 0.123 | 2.406 * | | | | | |
| | AQ: | | | | | | | | |
| | | Origin and ownership | 0.164 | 2.529 * | | | | | |

Note: ** $p < 0.001$; * $p < 0.05$.

**Table 7.** Results of Hierarchical Regression Analyses with SWB and AQ as Predictors of Participant MES in Deontology.

| Variable | | | β | t | F | R | R$^2$ | ΔR$^2$ | Adjusted R$^2$ |
|---|---|---|---|---|---|---|---|---|---|
| Deontology | | | | | | | | | |
| Step 1 | | | | | 19.967 ** | 0.270 | 0.073 | 0.073 | 0.069 |
| | Demographics | | | | | | | | |
| | | Monthly Salaries | 0.065 | 1.509 | | | | | |
| | | Religion | 0.258 | 6.028 ** | | | | | |

**Table 7.** *Cont.*

| Variable | | | β | t | F | R | R$^2$ | ΔR$^2$ | Adjusted R$^2$ |
|---|---|---|---|---|---|---|---|---|---|
| Step 2 | | | | | 62.889 ** | 0.521 | 0.272 | 0.199 | 0.267 |
| | Demographics | | | | | | | | |
| | | Monthly Salaries | 0.064 | 1.688 | | | | | |
| | | Religion | 0.111 | 2.768 * | | | | | |
| | SWB: | | | | | | | | |
| | Personal–communal | | 0.469 | 11.745 ** | | | | | |
| Step 3 | | | | | 49.916 ** | 0.532 | 0.283 | 0.012 | 0.278 |
| | Demographics | | | | | | | | |
| | | Monthly Salaries | 0.060 | 1.599 | | | | | |
| | | Religion | 0.107 | 2.681 * | | | | | |
| | SWB: | | | | | | | | |
| | Personal–communal | | 0.404 | 8.860 ** | | | | | |
| | Environmental | | 0.127 | 2.878 * | | | | | |
| Step 4 | | | | | 40.908 ** | 0.537 | 0.289 | 0.005 | 0.282 |
| | Demographics | | | | | | | | |
| | | Monthly Salaries | 0.058 | 1.528 | | | | | |
| | | Religion | 0.101 | 2.539 * | | | | | |
| | SWB: | | | | | | | | |
| | Personal–communal | | 0.389 | 8.433 ** | | | | | |
| | Environmental | | 0.107 | 2.372 * | | | | | |
| | Transcendental | | 0.080 | 1.944 | | | | | |
| Step 5 | | | | | 34.623 ** | 0.541 | 0.292 | 0.004 | 0.284 |
| | Demographics | | | | | | | | |
| | | Monthly Salaries | 0.062 | 1.650 | | | | | |
| | | Religion | 0.096 | 2.397 * | | | | | |
| | SWB: | | | | | | | | |
| | Personal–communal | | 0.366 | 7.580 ** | | | | | |
| | Environmental | | 0.106 | 2.346 * | | | | | |
| | Transcendental | | 0.081 | 1.977 * | | | | | |
| | AQ: | | | | | | | | |
| | Control | | 0.065 | 1.601 | | | | | |
| Step 6 | | | | | 34.948 ** | 0.572 | 0.328 | 0.035 | 0.318 |
| | Demographics | | | | | | | | |
| | | Monthly Salaries | 0.061 | 1.643 | | | | | |
| | | Religion | 0.058 | 1.461 | | | | | |
| | SWB: | | | | | | | | |
| | Personal–communal | | 0.181 | 3.049 * | | | | | |
| | Environmental | | 0.091 | 2.065 * | | | | | |
| | Transcendental | | 0.069 | 1.715 | | | | | |

**Table 7.** *Cont.*

| Variable | | β | t | F | R | $R^2$ | $\Delta R^2$ | Adjusted $R^2$ |
|---|---|---|---|---|---|---|---|---|
| | AQ: | | | | | | | |
| | Control | 0.055 | 1.386 | | | | | |
| | Origin and ownership | 0.287 | 5.139 ** | | | | | |
| Step 7 | | | | 30.850 ** | 0.574 | 0.330 | 0.002 | 0.319 |
| | Demographics | | | | | | | |
| | Monthly Salaries | 0.057 | 1.558 | | | | | |
| | Religion | 0.053 | 1.345 | | | | | |
| | SWB: | | | | | | | |
| | Personal–communal | 0.154 | 2.457 * | | | | | |
| | Environmental | 0.094 | 2.117 * | | | | | |
| | Transcendental | 0.067 | 1.656 | | | | | |
| | AQ: | | | | | | | |
| | Control | 0.046 | 1.135 | | | | | |
| | Origin and ownership | 0.295 | 5.253 ** | | | | | |
| | Reach | 0.056 | 1.335 | | | | | |
| Step 8 | | | | 46.427 ** | 0.575 | 0.331 | 0.001 | 0.319 |
| | Demographics | | | | | | | |
| | Monthly Salaries | 0.057 | 1.557 | | | | | |
| | Religion | 0.055 | 1.376 | | | | | |
| | SWB: | | | | | | | |
| | Personal–communal | 0.156 | 2.479 * | | | | | |
| | Environmental | 0.094 | 2.124 * | | | | | |
| | Transcendental | 0.064 | 1.569 | | | | | |
| | AQ: | | | | | | | |
| | Control | 0.057 | 1.286 | | | | | |
| | Origin and ownership | 0.301 | 5.276 ** | | | | | |
| | Reach | 0.055 | 1.304 | | | | | |
| | Endurance | −0.027 | −0.612 | | | | | |

Note: ** $p < 0.001$; * $p < 0.05$.

### 4.4. Moral Equity

The moral equity tendencies of accounting managers were analysed in eight steps (Table 3). In Step 1, demographic variables were used as control variables. The results showed that religion is a significant predictor of moral equity (β = 0.213, $t$ = 4.973, $p < 0.001$), explaining 8.4% of the variance (adjusted $R^2$ = 0.084). In Step 2, the personal–communal domain was added and subsequently determined to be a significant predictor (β = 0.598, $t$ = 16.286, $p < 0.001$), increasing the variance to 39.9% (adjusted $R^2$ = 0.399) with a significant change in $R^2$ ($\Delta R^2$ = 0.313); this finding can be classified to be of a medium effect size. Then, the environmental and transcendental domains were added in Steps 3 and 4, respectively; the findings showed that they are not significant predictors.

In Step 5, the control dimension of AQ was added and was subsequently determined to be a significant predictor (β = 0.120, $t$ = 3.214, $p < 0.05$), increasing the variance to 40.8% (adjusted $R^2$ = 0.408) with a significant change in $R^2$ ($\Delta R^2$ = 0.012). In Step 6, the origin and ownership dimension of AQ was added and was found to be a significant predictor

($\beta$ = 0.363, $t$ = 7.291, $p < 0.01$), increasing the variance to 49.6% (adjusted $R^2$ = 0.464) with a significant change in $R^2$ ($\Delta R^2$ = 0.056). In Step 7, the reach dimension of AQ was added and was determined to be a significant predictor ($\beta$ = 0.207, $t$ = 5.689, $p < 0.001$), increasing the variance to 49.6% (adjusted $R^2$ = 0.496) with a significant change in $R^2$ ($\Delta R^2$ = 0.032). In Step 8, the endurance dimension of AQ was added but was not established as a significant predictor of moral equity ($\beta$ = $-0.029$, $t$ = $-0.750$, $p > 0.05$).

### 4.5. Contractualism

Multiple regression was used to analyse in four steps the contractualism tendencies of accounting managers. In Step 1, the personal–communal domain was added to the model and was determined to be a significant predictor ($\beta$ = 0.198, $t$ = 4.545, $p < 0.001$), explaining 3.7% of the variance (adjusted $R^2$ = 0.037) with a significant change in $R^2$ ($\Delta R^2$ = 0.039). In Step 2, the environmental domain was added and was similarly established as a significant predictor ($\beta$ = 0.106, $t$ = 2.087, $p < 0.05$), increasing the variance to 4.4% (adjusted $R^2$ = 0.044), but the change in $R^2$ was not significant ($\Delta R^2$ = 0.008). In Step 3, the transcendental domain was added but was found to be a nonsignificant predictor ($\beta$ = 0.064, $t$ = 1.361, $p > 0.05$), i.e., a nonsignificant increase in variance to 4.5% (adjusted $R^2$ = 0.045) and a nonsignificant change in $R^2$ ($\Delta R^2$ = 0.003). In Step 4, the origin and ownership dimension of AQ was added and was found to be a significant predictor ($\beta$ = 0.235, $t$ = 3.662, $p < 0.001$), increasing the variance to 6.8% (adjusted $R^2$ = 0.068) with a significant change in $R^2$ ($\Delta R^2$ = 0.025). Lower (adjusted) R-squared values of up to 0.10 indicate that a substantial proportion of the variance in the dependent variable remains unexplained. However, in the social sciences and behavioural science, the focus is often on exploring relationships, trends and underlying mechanisms rather than high predictive accuracy [43].

### 4.6. Utilitarianism

The utilitarianism tendencies of accounting managers were analysed by multiple regression in two steps. In Step 1, the environmental domain was added to the model and was found to be a significant predictor ($\beta$ = 0.113, $t$ = 2.565, $p < 0.05$), explaining 1.1% of the variance (adjusted $R^2$ = 0.011) with a significant change in $R^2$ ($\Delta R^2$ = 0.013). In Step 2, the origin and ownership dimension of AQ was added to the model but was determined to be a nonsignificant predictor ($\beta$ = 0.067, $t$ = 1.364, $p > 0.05$), i.e., a nonsignificant increase in variance to 1.3% (adjusted $R^2$ = 0.013) and a nonsignificant change in $R^2$ ($\Delta R^2$ = 0.004).

### 4.7. Egoism

Multiple regression in three steps was used to analyse the egoism tendencies of accounting managers. In Step 1, the personal–communal domain was added to the model and was determined to be a significant predictor ($\beta$ = 0.148, $t$ = 3.364, $p < 0.05$), explaining 2.0% of the variance (adjusted $R^2$ = 0.020) with a significant change in $R^2$ ($\Delta R^2$ = 0.022). In Step 2, the environmental domain was added and was established as a significant predictor ($\beta$ = 0.134, $t$ = 2.623, $p < 0.05$), increasing the variance to 3.1% (adjusted $R^2$ = 0.031) with a significant change in $R^2$ ($\Delta R^2$ = 0.013). In Step 3, the origin and ownership dimension of AQ was added. The transcendental domain was found to be a significant predictor ($\beta$ = 0.164, $t$ = 2.529, $p < 0.05$), increasing the variance to 4.1% (adjusted $R^2$ = 0.041) with a significant change in $R^2$ ($\Delta R^2$ = 0.012).

### 4.8. Deontology

The deontology tendencies of accounting managers were analysed by multiple regression in eight steps. In Step 1, religion was found to be a significant predictor ($\beta$ = 0.258, $t$ = 6.028, $p < 0.001$), explaining 6.9% of the variance (adjusted $R^2$ = 0.069) of the deontology variable in MES. In Step 2, the personal–communal domain was added and was determined to be a significant predictor ($\beta$ = 0.469, $t$ = 11.745, $p < 0.001$), increasing the variance to 26.7% (adjusted $R^2$ = 0.267) with a significant change in $R^2$ ($\Delta R^2$ = 0.199).

In Step 3, the environmental domain was added and established as a significant predictor ($\beta = 0.127$, $t = 2.878$, $p < 0.05$), increasing the variance to 27.8% (adjusted $R^2 = 0.278$) with a significant change in $R^2$ ($\Delta R^2 = 0.012$). In Steps 4 and 5, the transcendental domain of SWB and the control dimension of AQ were added, but they were determined to be nonsignificant predictors. In Step 6, the origin and ownership dimension of AQ was added and was found to be a significant predictor ($\beta = 0.287$, $t = 5.139$, $p < 0.001$), increasing the variance to 31.8% (adjusted $R^2 = 0.318$) with a significant change in $R^2$ ($\Delta R^2 = 0.035$). In Steps 7 and 8, the reach and endurance dimensions of AQ were added but were found to be nonsignificant predictors.

Meanwhile, among the demographic factors, religion was the only significant predictor of moral equity and deontology. This finding indicates that the religious beliefs of people affect their EDM. The personal–communal domain consistently emerged as a significant predictor of contractualism, egoism and deontology; moral equity obtained a medium effect size ($\Delta R^2 > 0.3$). The environmental domain was a significant predictor of contractualism and utilitarianism, whereas the origin and ownership dimension was a significant predictor of moral equity and contractualism. However, the effect sizes for these predictors were lower than that of the personal–communal domain, indicating that they have a lower influence on EDM.

The results of hierarchical regression analysis partially support H2. Thus, each of the SWB domains and AQ dimensions can significantly predict the EDM tendencies of accounting managers in the CSR and financial management contexts. The overall finding highlights the need to integrate a wide range of factors when studying the EDM of accounting managers. In this study, the personal–communal domain emerged as a prominent predictor of many variables.

## 5. Discussion and Conclusions

The current study successfully addresses the research questions and hypothesis by examining the relationships between SWB, AQ, and EDM among accounting managers in the contexts of financial management and corporate social responsibility. The findings provide valuable insights into these relationships and offer unequivocal answers to the research questions.

Regarding Research Question 1, the findings of the study consistently support Hypothesis 1 (H1), which posits a positive correlation between SWB and ethical judgements for decision making. Notably, the personal–communal domain of SWB emerges as the most prominent predictor, exhibiting a significant positive correlation with moral equity, contractualism, egoism and deontology. These findings provide an answer to Research Question 3 and support Hypothesis 3 (H3). The personal–communal domain reflects a sense of fulfilment and the ability to connect with others. Particularly, the association between the personal–communal domain and moral justice demonstrates a medium effect size, indicating that accounting managers who experience fulfilment and strong connections with others are inclined to make moral choices guided by ideals of justice, fairness, and obligation. These findings align with previous research [28,44] that highlights the role of spirituality in ethical judgment.

Personal values, beliefs and attitudes are essential components of EDM. Personal moral beliefs (e.g., tolerance and optimism) are important determinants of moral decision making [45], whilst personal values and beliefs are major predictors of moral behaviour [46]. The current study finds that the environmental domain of SWB is a major predictor of contractarianism and utilitarianism. This result accords with the findings of study [47], in which the environmental components of SWB (e.g., harmony and peace manifesting in corporate culture and moral climate) are essential to moral decision making.

Regarding Research Question 2, the study consistently affirms Hypothesis 2 (H2), which suggests a positive association between AQ and ethical judgments for decision making. Significantly, the origin and ownership dimension of AQ emerges as the most prominent predictor, exhibiting a significant positive correlation with moral equity and

contractualism. These findings provide an answer to Research Question 4 and support Hypothesis 4 (H4). However, it is important to note that the effect sizes of both predictors are relatively low, indicating a less significant impact on EDM. The result obtained by this research is consistent with those provided by earlier studies [48–50] on the significant effects of personal traits on moral judgment. Furthermore, although the dispositional traits of moral identity and self-control are significant determinants of ethical conduct [48], emotional intelligence is also a significant predictor of EDM [49]. Individuals with high AQ are more likely to consider moral judgments, which may be associated with their ability to persevere amid hardship [16,50].

Religion is determined to be a significant predictor of moral equity and deontology. This result accords with the findings of other studies that established the significance of religious convictions in moral judgment. Religious convictions are a significant predictor of ethical conduct [51] and a significant predictor of EDM [52].

The current study establishes the importance of personal values, beliefs, attitudes, contextual circumstances and individual traits in predicting the EDM of accounting managers, contributing to the knowledge on EDM in the literature. The findings have important practical implications; they imply that initiatives and regulations that support social norms, individual attributes and environmental variables contribute to the successful fostering of ethical conduct at work. Future studies can build on these findings whilst addressing the shortcomings of the current study and utilising more rigorous research methodologies, as discussed in the succeeding sections.

Overall, this research highlights the need to consider multiple factors to explain the EDM of accounting managers. Here, the personal–communal domain consistently emerged as a strong predictor of many variables.

This study aims to investigate the associations and interactions of spirituality, AQ and EDM of accounting professionals in the financial management and CSR contexts. The research findings empirically support the hypotheses (H1 and H2) affirming that SWB and AQ are positively associated with ethical decision making. Furthermore, specific domains of SWB and dimensions of AQ are found to significantly predict the ethical judgments of accounting managers, thereby supporting the hypotheses (H3 and H4). These findings provide robust empirical evidence of the aforementioned relationships and highlight the significance of certain domains of spirituality and adversity quotients as significant predictors of ethical orientations among accountants. This research contributed to filling a research gap by shedding light on these overlooked relationships.

This study also makes a valuable conceptual contribution by integrating multiple theories to provide a holistic framework for examining the inner capacities and contextual factors that shape accountants' moral perspectives and behavior for these overlooked relationships. The research conclusions point to the importance of personal characteristics, contextual influence, and personal values, beliefs and attitudes in the EDM process. The personal–communal domain of SWB is a significant predictor of moral equity, contractualism, egoism and deontology, whereas the environmental domain of SWB is a significant predictor of contractualism, utilitarianism, egoism and deontology. Religion is determined to be a strong predictor of moral equity and deontology. The origin and ownership dimension of AQ is a significant predictor of moral equity, contractualism, egoism and deontology. The control and reach dimensions of AQ are a significant predictor of moral equity, respectively. These results underline the importance of integrating multiple variables in assessing the EDM of accounting managers.

Future research directions could be the investigation of the underlying mechanisms and processes that drive the inter-relationships between spirituality, SWB, AQ, and EDM in the field of accounting. Exploring mediating and moderating factors could lead to a more nuanced understanding of these relationships. In the future, longitudinal studies can be conducted to examine the long-term effects of spirituality, SWB, and AQ on accountants' ethical behavior. An evaluation of the effectiveness of interventions and training

programmes targeting spirituality, SWB, and AQ to promote ethical decision making would also be valuable.

## 6. Limitations and Suggestions for Future Studies

Despite the contributions of this research and the carefully considered methodology, several limitations need to be addressed. Firstly, as the study was restricted to Hong Kong respondents, its capacity to generalise the findings to other situations is constrained. Cultural and societal differences considerably affect the interrelations of spirituality, AQ and ethical judgment. Extreme care should be taken when applying the research findings to other areas.

Secondly, the study only used self-reported data, possibly leading to biases toward social desirability. Participants may not have expressed their genuine opinions and experiences but rather those scenarios they thought would be more socially acceptable. The precision and trustworthiness of the research conclusions may have also been constrained by the self-reporting nature of the data-gathering procedure.

Thirdly, the research conclusions appear to be inapplicable to other professions because only the context of accounting managers was examined. The associations and interactions of spirituality, AQ and EDM may vary depending on the diverse ethical norms practiced in other professions.

Finally, the cross-sectional methodology of this research entailed a complex evaluation of the interactions between spirituality, adversity tolerance and moral judgment. Establishing causation and determining the direction of the correlations between these variables requires longitudinal studies that can monitor the changes over time.

The aforementioned limitations should be considered by future researchers. Studies on the association between spirituality, AQ and EDM in the financial management and CSR contexts can be replicated in other contexts or professions.

**Author Contributions:** H.-K.P. led the data collection and analysis and the writing of the original draft. C.-C.F. led the funding acquisition and the writing of the original draft and its review and editing. All authors have read and agreed to the published version of the manuscript.

**Funding:** This research is supported by the research project funding of Corporate Social Responsibility: Perceptions and Activities (RP/FCG-01/2022) from Macao Polytechnic University.

**Institutional Review Board Statement:** Not applicable.

**Informed Consent Statement:** Informed consent was obtained from all subjects involved in the study.

**Data Availability Statement:** All data used in this evaluation are available and can be requested from the authors. Please contact Hok Ko Pong (author, hkpong@thei.edu.hk) for data requests.

**Conflicts of Interest:** The authors declare no conflict of interest.

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
