# Peer review of "The Associations of Spirituality, Adversity Quotient and Ethical Decision Making of Accounting Managers in the Contexts of Financial Management and Corporate Social Responsibility"

_sustainability, doi:10.3390/su151914287_

Round 1
Reviewer 1 Report
The paper is well structured with significant contributions to the current body of knowledge. Research subject and problem have been defined correctly and are interesting in terms of profiling research in this area. Specific objectives and research tasks are related to it. The content of the article corresponds to the scientific problem - substantive compatibility in theoretical, methodological, and empirical terms. The research methodology used in the paper is correct.The results of the analysis are described correctly. This applies both to conclusions on quantitative results and to qualitative implications searched for.
The following suggestions can be considered:
• The abstract could be improved by incorporating the main objective of the research. The main goal has not been formulated in the abstract or in the introduction.
• In ‘Conclusions’ the authors should clarify the conceptual and empirical contribution of the paper. Also, there should be clearly written what are further research in this area.
Formal requirements (language, edition, etc.): In general, the language is stylistically and formally correct, standard proofreading is necessary. Bibliographic sources are not very extensive, I suggest expanding them, to better serve (number and structure) to achieve the objectives of the research.
Author Response
Dear anonymous reviewer(s),
Thank you very much for taking the time to review my work and for providing me with valuable comments and constructive suggestions. I greatly appreciate your efforts to improve the quality of my manuscript. I have carefully reviewed each of your comments and revised them accordingly. All the revised works are provided with track changes. We have really learnt a lot from the revisions. Below I have listed the reviewers' comments and the corresponding revisions I have made:
|
Comments from Reviewer 1 |
|
Responses from Authors |
|
The abstract could be improved by incorporating the main objective of the research. The main goal has not been formulated in the abstract or in the introduction. |
|
We agreed and revised. The objective and the aim of the study are mentioned in Abstract. Also, the main goals of the study are mentioned in the section of “Introduction”. |
|
• In ‘Conclusions’ the authors should clarify the conceptual and empirical contribution of the paper. Also, there should be clearly written what are further research in this area. |
|
We agreed and revised. The conceptual and empirical contribution (i.e. empirical evidence) as well as the further research in this area are mentioned in the section of “Conclusion”. |
|
|
|
|
Thank you very much!
Best Regards,
Authors
Reviewer 2 Report
Thank you for giving me an opportunity to read this manuscript. I am recommending few of the suggestions to further improve the manuscript.
Though methods section is the strength of this manuscript, however, there are notable deficiencies in the structure of the introduction, hypothesis development, and it's alignment with the subsequent discussion. Addressing the shortcomings related to the theoretical contribution, introduction, hypothesis development, and their integration with the discussion is essential for enhancing the clarity, depth, and academic rigor of the manuscript. This will enable readers to grasp the study's purpose, methodologies, and findings in a more cohesive and meaningful manner.
Good Luck
Checked relevant box above
Author Response
Dear anonymous reviewer(s),
Thank you very much for taking the time to review my work and for providing me with valuable comments and constructive suggestions. I greatly appreciate your efforts to improve the quality of my manuscript. I have carefully reviewed each of your comments and revised them accordingly. All the revised works are provided with track changes. We have really learnt a lot from the revisions. Below I have listed the reviewers' comments and the corresponding revisions I have made:
|
Comments from Reviewer 2 |
|
Responses from Authors |
|
There are notable deficiencies in the structure of the introduction, hypothesis development, and it's alignment with the subsequent discussion.
Addressing the shortcomings related to the theoretical contribution, introduction, hypothesis development, and their integration with the discussion is essential for enhancing the clarity, depth, and academic rigor of the manuscript. This will enable readers to grasp the study's purpose, methodologies, and findings in a more cohesive and meaningful manner. |
|
We agreed and revised in the section of introduction, including hypothesis development and discussion. We have remedied the shortcomings pointed out in the structure of the introduction, the development of the hypotheses and their coordination with the subsequent discussion. |
|
|
|
|
Reviewer 3 Report
The peer-reviewed scientific study of up to 21 pages is undoubtedly a very interesting scientific work, the content of which can be beneficial not only for theory but also for practice.
However, the authors probably neglected to sufficiently read the instructions for authors available on the journal's website and strictly follow them so that the content structure is respected.
For this reason, the deficiency already occurs in the abstract, where I recommend expanding it by setting the goal, the scientific research methods used (one can only be deduced) as well as specifying the results of the study themselves.
The authors seem to be confusing terms like "keywords" and "keyphrases" because the last "keyword" consists of up to six words
The introduction lacks a really clearly defined main goal and secondary goals. I found four established hypotheses as well as four research questions that the authors should have clearly answered in the final chapter, thus fulfilling the very meaning of their scientific work. These unequivocal answers are missing.
The relatively short introduction lacks a definition of the importance of the topic as well as an explanation of the basic term "manager". It is necessary to thoroughly explain who he is, how he is related to the organization, how he does it, for example:
Peráček T. & Kaššaj M. (2023). The influence of jurisprudence on the formation of relations between the manager and the limited liability company. Juridical Tribune. 13 (1), pp. 43-62. doi: 10.24818/TBJ/2023/13/1.04
The number of used sources (42) for the number of pages of the scientific study (21) is insufficient. To increase the scientific value of the work, I recommend the authors to expand this number by, for example, works:
Tache, CEP and Sararu, CS. 2023. NEW TRANSDISCIPLINARY DIRECTIONS IN INTERNATIONAL LAW? LEX HUMANA, 15 (4), pp. 86-109
Mucha, B.: Evaluation of the State of Implementation of the European Structural and Investment Funds: Case Study of the Slovak Republic. Online Journal Modeling the New Europe, 35:4-24, (2021), doi: 10.24193/OJMNE.2021.35.0
As I mentioned, in the conclusion it is necessary to clearly answer the established research questions and hypotheses with proper justification. In the end, I recommend devoting a few sentences to the question of further research.
Author Response
Dear anonymous reviewer(s),
Thank you very much for taking the time to review my work and for providing me with valuable comments and constructive suggestions. I greatly appreciate your efforts to improve the quality of my manuscript. I have carefully reviewed each of your comments and revised them accordingly. All the revised works are provided with track changes. We have really learnt a lot from the revisions. Below I have listed the reviewers' comments and the corresponding revisions I have made:
|
Comments from Reviewer 3 |
|
Responses from Authors |
|
The authors probably neglected to sufficiently read the instructions for authors available on the journal's website and strictly follow them so that the content structure is respected.
For this reason, the deficiency already occurs in the abstract, where I recommend expanding it by setting the goal, the scientific research methods used (one can only be deduced) as well as specifying the results of the study themselves.
|
|
We agreed and revised. The objective and the aim of the study are mentioned in Abstract. Also, the main goals of the study are mentioned in the section of “Introduction”. The headings of Research backgrounds, Research objectives and Research questions are clearly mentioned in the section of “Introduction”. |
|
The authors seem to be confusing terms like "keywords" and "keyphrases" because the last "keyword" consists of up to six words.
|
|
We agreed and revised. They are divided into financial management; corporate social responsibility. |
|
The introduction lacks a really clearly defined main goal and secondary goals. I found four established hypotheses as well as four research questions that the authors should have clearly answered in the final chapter, thus fulfilling the very meaning of their scientific work. These unequivocal answers are missing.
|
|
We agreed and revised. The main goals of the study are mentioned in the section of “Introduction”. Also, we revised the final chapter (the section of “Discussion” and “Conclusions”) to answer the questions and hypothesis of the study |
|
The relatively short introduction lacks a definition of the importance of the topic as well as an explanation of the basic term "manager". It is necessary to thoroughly explain who he is, how he is related to the organization, how he does it.
|
|
We agreed and revised. We added paragraphs to define the term of “manager” and the importance of accounting managers to organization with recent studies in the section of “introduction”. |
|
The number of used sources (42) for the number of pages of the scientific study (21) is insufficient.
|
|
We agreed and revised. The numbers of used sources now have been increased to 52. |
|
As I mentioned, in the conclusion it is necessary to clearly answer the established research questions and hypotheses with proper justification. In the end, I recommend devoting a few sentences to the question of further research.
|
|
We agreed and revised. Now it is better to clearly and directly answer the research questions and hypotheses with proper justification. Also, the further research in this area is included. |
|
|
|
|
Reviewer 4 Report
Congratulations, your paper is very interesting and very much relevant for the literature on CSR and related topics.
Nevertheless, your adjusted r-squared figures are very low. This may be considered OK, if you justify it with the nature of your work (social sciences, human behavior).
Look for literature related to this issue, may authors justify up to 0.10 figures.
Nothing relevant found
Author Response
Dear anonymous reviewer(s),
Thank you very much for taking the time to review my work and for providing me with valuable comments and constructive suggestions. I greatly appreciate your efforts to improve the quality of my manuscript. I have carefully reviewed each of your comments and revised them accordingly. All the revised works are provided with track changes. We have really learnt a lot from the revisions. Below I have listed the reviewers' comments and the corresponding revisions I have made:
|
Comments from Reviewer 4 |
|
Responses from Authors |
|
Nevertheless, your adjusted r-squared figures are very low. This may be considered OK, if you justify it with the nature of your work (social sciences, human behavior). Look for literature related to this issue, may authors justify up to 0.10 figures. |
|
We agreed and revised. We added a paragraph with the recent study of Ozili, P. K. (2023) to justify lower (adjusted) R-squared value. “Lower (adjusted) R-squared values of up to 0.10 indicate that a substantial proportion of the variance in the dependent variable remains unexplained (Saliya, 2023). However, in the social sciences and behavioural science, the focus is often on exploring relationships, trends and underlying mechanisms rather than high predictive accuracy (Ozili, P. K. (2023).”
|
|
|
|
|
Thank you very much!
Best Regards,
Authors
Round 2
Reviewer 2 Report
Upon reviewing the introductory section, abundant with its numerous headings, there arises a sense that this is more akin to a research report rather than a concise research article. It is recommended to peruse the introductions of various research articles and formulate the content in a more polished and streamlined fashion.
Selected above.
Author Response
We would like to express my heartfelt gratitude for your diligent review of our work and for sharing your valuable comments and constructive suggestions. Your time and effort in providing feedback have been immensely valuable in enhancing the quality of our manuscript. We would like to acknowledge your commitment to fostering academic growth and knowledge advancement. I have thoroughly examined each of your comments and diligently incorporated the suggested revisions, which are clearly indicated with track changes. We have learned a great deal from your expertise. Once again, we extend my sincere appreciation for your invaluable contribution to the improvement of our work.
|
Comments from Reviewer 2 |
|
Responses from Authors |
|
Upon reviewing the introductory section, abundant with its numerous headings, there arises a sense that this is more akin to a research report rather than a concise research article. It is recommended to peruse the introductions of various research articles and formulate the content in a more polished and streamlined fashion. |
|
Yes. We agreed. We have reviewed and made revisions in accordance with the instructions provided, taking into consideration several research articles published in the journal.
|
|
|
|
|
Reviewer 3 Report
Dear Authors,
after assessing the revisions made, it can be concluded that you have met most of the recommendations. What worries me, however, is the misunderstanding of the need to define the position of a manager, not from an economic point of view, but from a legislative point of view. The reason is the fact that only the legislative point of view is binding and ultimately legally enforceable. So the opinions of economists can be inspiring but not binding.
It is also strange why the chapter "Limitation" comes before the chapter "Conclusion" Please explain and complete the paper according to the instructions.
Author Response
We would like to express my heartfelt gratitude for your diligent review of our work and for sharing your valuable comments and constructive suggestions. Your time and effort in providing feedback have been immensely valuable in enhancing the quality of our manuscript. We would like to acknowledge your commitment to fostering academic growth and knowledge advancement. We have thoroughly examined each of your comments and diligently incorporated the suggested revisions, which are clearly indicated with track changes. We have learned a great deal from your expertise. Once again, we extend my sincere appreciation for your invaluable contribution to the improvement of our work.
|
Comments from Reviewer 3 |
|
Responses from Authors |
|
after assessing the revisions made, it can be concluded that you have met most of the recommendations. What worries me, however, is the misunderstanding of the need to define the position of a manager, not from an economic point of view, but from a legislative point of view. The reason is the fact that only the legislative point of view is binding and ultimately legally enforceable. So the opinions of economists can be inspiring but not binding. |
|
Yes. We agreed. We have reviewed and made revisions the definition of managers, particularly accounting managers (the research topic) from the legislative point of view. |
|
It is also strange why the chapter "Limitation" comes before the chapter "Conclusion" Please explain and complete the paper according to the instructions. |
|
Yes. We agreed. We have reviewed and made revisions in accordance with the instructions provided, taking into consideration several research articles published in the journal. |
Round 3
Reviewer 2 Report
Thank you for incorporating changes. However,
1] How come you write the hypotheses at the end of introduction section whn you have already a separate heading of Literature Review? please, insert your study hypotheses at the appropriate place.
2] As the authors have done a very concise discussion, my recommendation is to merge the heading of discussion and conclusion.
Mentioned.
Author Response
Thank you very much for giving us the comments again.
|
Comments from Reviewer 2 |
|
Responses from Authors |
|
1] How come you write the hypotheses at the end of introduction section whn you have already a separate heading of Literature Review? please, insert your study hypotheses at the appropriate place. |
|
Yes. We agreed. We have reviewed and made revisions in accordance with the instructions provided, taking into consideration several research articles published in the journal.
|
|
2] As the authors have done a very concise discussion, my recommendation is to merge the heading of discussion and conclusion.
|
|
Yes. We agreed. We merged the heading of “Discussion and Conclusion”. |
Reviewer 3 Report
I agreee to publish
Author Response
Thank you very much for accepting our publication.